# Toxicological Comparison of Mancozeb and Zoxamide Fungicides at Environmentally Relevant Concentrations by an In Vitro Approach

**DOI:** 10.3390/ijerph18168591

**Published:** 2021-08-14

**Authors:** Gabriele Lori, Roberta Tassinari, Laura Narciso, Ion Udroiu, Antonella Sgura, Francesca Maranghi, Sabrina Tait

**Affiliations:** 1Gender-Specific Prevention and Health Unit, Center for Gender-Specific Medicine, Istituto Superiore di Sanità, Viale Regina Elena 299, 00161 Rome, Italy; gabriele.lori@iss.it (G.L.); roberta.tassinari@iss.it (R.T.); laura.narciso@iss.it (L.N.); sabrina.tait@iss.it (S.T.); 2Science Department, Università Degli Studi di Roma Tre, Viale Guglielmo Marconi 446, 00146 Rome, Italy; ion.udroiu@uniroma3.it (I.U.); antonella.sgura@uniroma3.it (A.S.)

**Keywords:** gene expression, genotoxicity, pesticides, reactive oxygen species, risk assessment

## Abstract

Mancozeb (MZ) and zoxamide (ZOX) are fungicides commonly used in pest control programs to protect vineyards. Their toxic and genotoxic potential were investigated in vitro on HepG2 and A549 cell lines at environmentally relevant concentrations. Cytotoxicity, apoptosis, necrosis and intracellular reactive oxygen species (ROS), comet assay and a micronucleus test with CREST immunofluorescence were used. The expression of a panel of genes involved in apoptosis/necrosis (*BAX/BCL2*), oxidative stress (*NRF2*), drug metabolism (*CYP1A1*) and DNA repair (*ERCC1/OGG1*) was evaluated by real-time PCR. Both fungicides were cytotoxic at the highest tested concentrations (295.7 and 463.4 µM, respectively); MZ induced necrosis, ZOX did not increase apoptosis but modulated *BAX* and *BCL2* expression, suggesting a different mechanism. Both compounds did not increase ROS, but the induction of *CYP1A1* and *NRF2* expression supported a pro-oxidant mechanism. The comet assay evidenced MZ genotoxicity, whereas no DNA damage due to ZOX treatment was observed. Positive micronuclei were increased in both cell lines treated with MZ and ZOX, supporting their aneugenic potential. *ERCC1* and *OGG1* were differently modulated, indicating the efficient activation of the nucleotide excision repair system by both fungicides and the inhibition of the base excision repair system by MZ. Overall, MZ confirmed its toxicity and new ZOX-relevant effects were highlighted.

## 1. Introduction

The sustainable use of pesticides is currently requested by the Directive 2009/128/EC, which addresses the reduction of the risks and the impacts of pesticide on human health and the environment through integrated pest management and the use of alternative approaches (e.g., reduction, substitution or non-chemical alternatives to pesticides). In Italy, on the basis of the Directive, different agronomical protocols to treat and protect vineyards from seasonal infections of fungi and molds, which are also frequently associated with climate changes, are used [1]. Even though agrochemicals should be extremely specific to their relative target organisms due to the mode of action, they are not always completely selective. This represents a risk for all non-target species, including humans [2]. This has prompted agronomists to implement customized agronomical protocols based, whenever possible, on agrochemicals with a lower impact on human and environmental health to replace substances identified as higher risk. As an example, mancozeb (MZ) and zoxamide (ZOX) are fungicides used to treat two of the most common grapevine diseases caused by fungal pathogens, namely, downy mildew (Peronospora-Plasmopara viticola) and powdery mildew (Oidium-Uncinula necator, Oidium tuckeri). MZ is a compound still widely used in traditional protocols, whereas ZOX has been recently introduced as a potentially less toxic MZ substitute. MZ belongs to the ethylene-bis dithiocarbamate family (EBDTCs) with multi-site activity; ethylene thiourea (ETU)—its main active metabolite—exerts carcinogen, teratogen and goitrogen effects [3,4]. In mammals, MZ and ETU have been detected in several biological compartments, with the highest residues found in thyroid where they cause the inhibition of thyroid peroxidase and hyperplasia/hypertrophy, as observed in short and long term toxicity in vivo studies [5,6]. Beyond the thyroid, MZ displays other endocrine-disrupting activities that induce developmental and reproductive effects, that are mainly mediated by thyroid homeostasis or by oxidative and genotoxic mechanisms [7]. Recent in vitro data show that MZ also exerts toxic effects on liver cells, leading to cell death [8]. As for its reproductive effects, the mechanism may be attributed to mitochondrial-mediated apoptosis through reactive oxygen species (ROS) generation [9,10], or to DNA damage, as observed in several in vitro and in vivo studies [11,12,13,14]. MZ is also a suspected carcinogen since it increases the incidence of hepatocarcinoma in male Wistar rats at a 1000 ppm dose level [15]. Despite its known toxic properties, MZ is still largely present in several protocols worldwide, although in the last few years its use has been somewhat limited.

ZOX belongs to the benzamide family and it is known to cause mitotic arrest by specifically binding to β-tubulin, inhibiting tubuli polymerization, and consequently, cell division [16,17]. Recently, due to its mode of action and high anti-resistance properties, ZOX has been included in several protocols to treat vineyards against Peronospora. The liver is the main target organ and it displays increased weight and hypertrophy in both short and long term in vivo mammalian toxicity studies [18]. In mice models, ZOX did not exert clastogenic, aneugenic or genotoxic effects, whereas aneuploidy induction was observed in vitro [19].

In this framework, the aim of the present study was to develop an in vitro testing approach to compare the two fungicides from the toxicological point of view and to evaluate the actual validity of MZ substitution in agronomical protocols. Hepatocellular carcinoma (HepG2) and human lung carcinoma epithelial cell lines (A549) were selected as representative of the main target organs, particularly for professional exposure. Both cell lines are metabolically competent; thus, they represent a valuable model for in vitro testing of chemicals [20,21,22,23]. Although both HepG2 and A549 are cancer cell lines, they were selected because of their wide use in the scientific community, especially in toxicological studies, allowing a broader comparison of the results with those available in the literature.

The approach involved the evaluation of cytotoxicity, apoptosis, necrosis, oxidative stress, and genotoxicity as well as analysis of the gene expression of a panel of genes involved in apoptosis/necrosis regulation (*BAX* and *BCL2*), oxidative stress response (*NRF2*), drug metabolism (*CYP1A1*) and DNA repair systems (*ERCC1*, *OGG1*).

## 2. Materials and Methods

### 2.1. Cell Lines and Chemicals

Hepatocellular carcinoma cells HepG2 [HEPG2] (ATCC HB-8065) and human lung carcinoma epithelial cell line A549 (ATCC CCL-185) were purchased from American Type Culture Collection (ATCC, Manassas, VI, USA). Both cell lines were cultured in DMEM without phenol red (Gibco, Milan, Italy), supplemented with 10% fetal bovine serum (Gibco), 100 U/mL penicillin, 100 µg/mL streptomycin (Gibco) and 2 mM L-Glutamine (Gibco). Cells were grown in an incubator at 37 °C, 5% CO_2_ and 90% humidity.

Mancozeb (MZ, CAS no. 8018-01-7, purity 99%) and zoxamide (ZOX, CAS no. 156052-68-5, purity 99%) were purchased from Sigma-Aldrich (Milan, Italy), and dissolved in sterile DMSO (Sigma-Aldrich) to obtain stock solutions of 36.9 mM and 231.7 mM, respectively.

Just before use, serial dilutions of chemicals in culture medium were prepared considering the concentration used in the field by agricultural workers as 1X (starting point). For MZ, due to its limited solubility, the maximum dilution required to obtain an acceptable DMSO concentration (0.8%) corresponded to 1/10 of the field concentration (0.1X).

### 2.2. Cytotoxicity Assays

HepG2 and A549 cell proliferation and viability were assessed by CyQUANT (CyQUANT^®^ Direct Cell Proliferation Assay; Life Technologies, Paisley, UK) and MTS (CellTiter 96^®^ AQueous One Solution reagent; Promega, Madison, WI, USA) assays, respectively, following the manufacturers’ protocols. For each assay, 10,000 cells diluted in 100 µL culture medium/well were plated on 96 flat-bottomed multi-wells and incubated overnight at 37 °C to allow adhesion. Medium was replaced with fresh medium containing six ten-fold serial dilutions of MZ starting from 0.1X field concentration (295.7–0.002957 μM), ZOX starting from 1X field concentration (463.4–0.004634 μM), or vehicle at the percentage corresponding to the highest concentration tested for the two chemicals (0.8% and 0.2% for MZ and ZOX, respectively) as control. All treatments were performed in triplicate, and plates were incubated for 24 h at 37 °C. At the end, 100 µL 2X CyQuant Detection Reagent or 20 μL MTS reagent was added to each well, and incubated for 1 h at 37 °C. By using the Victor 3 Multilabel Reader (PerkinElmer, MA, USA), fluorescence was read from the bottom with a green filter (485 nm excitation, 535 nm emission) and an absorbance of 490 nm for the CyQuant and MTS assays. The vehicle control cells were set as 100% viable. Each assay was repeated in three independent experiments.

Dose–response curves were derived and visualized by GraphPad Prism v5.01 (GraphPad Software Inc., La Jolla, CA, USA) whereas EC_10_, EC_20_ and EC_50_ values were calculated by the *drc* v3.0-1 package in R 4.0.4 using the four-parameter log-logistic function [24].

### 2.3. Apoptosis–Necrosis Assay

The RealTime-Glo™ Annexin V Apoptosis and Necrosis Assay kit (Promega) was used following the manufacturer’s protocol. For both HepG2 and A549, 10,000 cells in 100 µL/well were seeded on 96 white flat-bottomed multi-wells, and incubated overnight at 37 °C to allow adhesion. After 24 h, the medium was removed and cells were treated with the three highest concentrations of MZ (at 295.7, 29.6 and 2.96 μM) or ZOX (at 463.4, 46.34 and 4.63 μM), or vehicle as control (DMSO 0.8% and 0.2%) in duplicated wells, and each being concentrated two-fold. Then, 100 μL/well of reagent mix solution was added to reach nominal concentrations. Plates were incubated at 37 °C and they were read each hour for the first 8 h and then at 24 h for both luminescence and fluorescence (485/535 nm) by a Victor 3 Multilabel Reader (PerkinElmer), detecting apoptosis and necrosis signals, respectively. The assay was repeated in three independent experiments.

### 2.4. Reactive Oxygen Species (ROS) Assay

The ROS Detection Assay Kit (BioVision, Milpitas, CA, USA) was used to measure the amount of intracellular ROS following the manufacturer’s protocol. Briefly, 10,000 HepG2 or A549 cells in 100 µL of culture medium/well were seeded on 96 flat-bottomed multi-wells and incubated overnight at 37 °C to allow adhesion. After 24 h, the medium was removed and cells were washed once with 100 μL ROS assay buffer, combined with 100 μL/well 1X ROS assay label, and incubated for 1 h at 37 °C. At the end, the ROS label solution was removed and cells were treated with the three highest concentrations of MZ (at 295.7, 29.6 and 2.96 μM), ZOX (at 463.4, 46.34 and 4.63 μM), vehicle as negative control (DMSO 0.8% and 0.2% for cells treated with MZ and ZOX, respectively), or H_2_O_2_ 100 μM as positive control in duplicated wells, and incubated for 24 h at 37 °C. Plates were read for green fluorescence from the bottom (485 nm excitation, 535 nm emission) by the Victor 3 Multilabel Reader (PerkinElmer). Three independent experiments were performed. After background subtraction, the fold-change of the fluorescence reading of treated cells with respect to vehicle control cells was calculated.

### 2.5. Alkaline Comet Assay

The alkaline single cell gel electrophoresis (SCGE) was performed as described by Singh et al. [25] with slight modifications. Briefly, 300,000 cells/2 mL culture medium were seeded on 35 mm dishes and incubated overnight at 37 °C for adhesion. After 24 h, cells were treated with the two fungicides at concentrations derived by cytotoxicity assays and ranging from maximum acceptable cytotoxic concentration (about 50–70% viability) to lower or non-cytotoxic concentrations (≥70% viability), as recommended by the guidelines for in vitro genetic toxicology testing [26]. The concentrations of MZ were 29.6 μM, 2.96 μM and 296 nM; the concentrations of ZOX were 46.3 μM, 4.63 μM and 463 nM. Cells were treated in triplicate for 5 h or 24 h at 37 °C including medium control cells. Treatment with methyl methane sulfonate (MMS) 1 mM for 30 min was performed as positive control. At the end of incubation, cells were washed with phosphate buffered saline (PBS) solution (KCl 2.6 mM, KH_2_PO_4_ 1.47 mM, NaCl 137.9 mM and Na_2_HPO_4_-7H_2_O 8 mM, without calcium and magnesium), trypsinized (Trypsin 0.25% without phenol red; Gibco), harvested and centrifuged at 261× *g* for 8 min at room temperature (RT), then the pellet was resuspended in 30 μL PBS. Cell suspension was mixed with 195 μL 0.7% low melting agarose. For each sample, two slides were prepared quickly dropping 75 μL of the volume onto each previously coated slide and air dried with 1% normal melting agarose, covered with a glass coverslip and placed at +4 °C for 10 min. After removing the coverslips, the slides were left submerged in a cold lysis solution (Na_2_EDTA 100 mM, NaCl 2.5 M, Tris 10 mM) overnight and then placed in the electrophoresis chamber in the dark, which contained a cold alkaline buffer solution (NaOH 300 mM, Na_2_EDTA 1 mM, final pH ≥ 13). After DNA denaturation (20 min at +4 °C), and electrophoresis (25 V, 300 mA, 20 min), the slides were neutralized (Tris 400 mM in H_2_O, pH 7.5), fixed in cold methanol and air dried. Slides were stained with 50 µL Gel Red (1 µL/mL in PBS) and analyzed by a fluorescence microscope (Nikon ECLIPSE 80i) at 20X magnification. Around 75 not overlapped and randomly chosen nucleoids from each of the two slides were examined by an image analysis system (LUCIA Comet Assay™, Laboratory Imaging, Prague, Czech Republic) that recorded the percentage of tail intensity (% TI). The assay was repeated in three independent experiments.

### 2.6. Cytokinesis-Block Micronucleus Assay (CBMN) and CREST Immunofluorescence

The cytokinesis-block micronucleus assay (CBMN) was performed as described by Fenech [27] with a few modifications. For both HepG2 and A549, 200,000 cells/2 mL culture medium were seeded on 35 mm dishes and incubated overnight at 37 °C to allow adhesion. Cells were treated for 24 h with MZ (29.6 μM; 14.79 μM; 2.96 μM; 296 nM; 29.6 nM and 2.96 nM), ZOX (9.27 μM; 5.79 μM; 4.63 μM; 2.32 μM and 1.16 μM) or medium alone as control, in the presence of Cytochalasin-B (Cyt-B, 3 µg/mL in DMSO; Sigma Aldrich, St. Louis, MO, USA), an inhibitor of the actin microfilament ring assembly. Colchicine 10 ng/mL (Sigma Aldrich, St. Louis, USA) was used as positive control. The maximum concentrations used were selected on the basis of cytotoxicity data, avoiding concentrations causing the 55 ± 5% cell death, as recommended by OECD [28], as well as on the percentage of binucleated cells (% BNC), which was always higher than 50% of the total cell number (data not shown). After the treatment, the medium was removed, cells were washed with PBS, trypsinized, harvested and centrifuged at 4 °C, 261× *g*, for 8 min. Cell pellets were resuspended in 500 μL PBS and 80–100 µL of suspension was put in a cytospin cuvette and centrifuged by the cytospin at 400 rpm for 5 min (Thermo Shandon Cytospin 3; Pittsburgh, PA, USA) and fixed for 30 min on slides in cold methanol (−20 °C). Slides were then washed in PBS and a blocking solution (Bovine Serum Albumin, BSA 1%) was added. After 30 min, the BSA was removed and 15 μL of primary antibody (CREST anti-kinetochore-Human diluted 1:1 in BSA 1%, Antibodies Incorporated, Davis, CA, USA) was added to each slide, which were covered with parafilm and incubated in a humidified chamber for 1 h at 37 °C. Slides were extensively washed with BSA 1% and PBS before adding the secondary antibody (FITC Rabbit anti-Human diluted 1:80 in BSA 1%, Sigma-Aldrich). The incubation and wash steps were repeated, slides were stained with 15 µL of a 1:1 solution of DAPI (final concentration 1 µg/mL; Sigma Aldrich, St. Louis, MO, USA)/antifade (Vectashield, Vector Laboratories, Burlingame, CA, USA). For each experimental point, the frequency of micronuclei (MNi) in 500 BN cells was assessed by fluorescence microscope reading (Axio Imager M1 equipped with a CCD camera, Carl Zeiss, Berlin, Germany), distinguishing between clastogenic, i.e., CREST-negative, (MN−) and aneugenic, i.e., CREST-positive, damage (MN+). The experiments were repeated three times independently.

### 2.7. Gene Expression Analysis

HepG2 and A549 cells were plated with 300,000 cells in 1.5 mL/well, in 6 flat-bottomed multi-wells and allowed to adhere overnight at 37 °C. After 24 h, the medium was removed, and cells were treated with three non-cytotoxic concentrations of MZ (29.6 μM–296 nM), ZOX (4.63 μM–46.34 nM), or medium alone as control for 24 h. At the end, cells were washed with PBS, trypsinized, harvested and centrifuged at 261× *g* for 5 min at RT. The supernatants were removed and cell pellets were stored at −80 °C until analysis. Three independent experiments were performed. Total RNA was extracted from each sample using the Norgen RNA kit (Norgen, Thorold, ON, Canada) according to the manufacturer’s protocol. RNA quantity was assessed with Nabi Nano Spectrophotometer (MicroDigital Co. Ltd., Seongnam-si, Korea) and then 1 μg of total RNA from each sample was reverse transcribed to cDNA using the SensiFast™ cDNA Synthesis Kit (Bioline Reagents Ltd., London, UK) according to the manufacturer’s instructions. Specific primers for BCL2 apoptosis regulator (*BCL2*), BCL2 associated X apoptosis regulator (*BAX*), nuclear factor erythroid 2 like 2 (*NRF2*), cytochrome P450 family 1 subfamily A member 1 (*CYP1A1*), 8-oxoguanine DNA glycosylase (*OGG1*) and ERCC excision repair 1, endonuclease non-catalytic subunit (*ERCC1*), as well as for glyceraldehyde-3-phosphate dehydrogenase (*GAPDH*) as the reference gene, were designed using the Primer-BLAST web application (www.ncbi.nlm.nih.gov/tools/primer-blast, last access 9 November 2020) and purchased from Invitrogen (Thermo Fisher Scientific). Primers sequences are listed in Table 1. The SensiMix SYBR kit (Bioline) was used to perform real-time PCR assays, running reactions in duplicate on a Bioer LineGene 9600 Plus thermocycler instrument (Bioer Technology Co. Ltd., Hangzhou, China). The thermal program was as follows: 1 cycle at 95 °C for 10 min; 40 cycles at 95 °C for 15 s, 58 °C for 30 s and 72 °C for 1 min; 1 dissociation cycle from 55 to 95 °C, 30 s/°C was added to verify the amplification products. The LineGene 9600 PCR V.1.0 software (Bioer) was used to determine the threshold cycles (Ct). ΔΔCt values were calculated using vehicle control cells as calibrators and *GAPDH* as normalizer gene.

### 2.8. Statistical Analysis

All experimental data were analyzed with JMP 10 software (SAS Institute Inc., Cary, NC, USA) by one-way analysis of variance (ANOVA). Where applicable, post-hoc pairwise comparisons among treated and control groups were performed by the Dunnett’s test. A *p*-value < 0.05 was considered statistically significant.

## 3. Results

### 3.1. Cytotoxicity

In HepG2 cells, MZ strongly decreased cell proliferation and vitality by more than 83% at 295.7 μM—the highest concentration tested—and also significantly reduced cell vitality at 29.6 μM (Figure 1). ZOX induced a significant decrease in cell proliferation and viability at the two highest concentrations, 463.4 μM and 46.34 μM, with a more marked effect on proliferation. As shown in Table 2, the corresponding EC_10_, EC_20_ and EC_50_ were lower for MZ as regards vitality and lower for ZOX in relation to proliferation.

In A549 cells, both compounds significantly reduced cell proliferation by about 50% at the highest concentrations. In addition, MZ significantly decreased cell vitality at the two highest concentrations tested (295.7 μM and 29.6 μM) (Figure 1), whereas ZOX did not exert any significant effect on cell vitality. For both assays, lower EC_x_ values were obtained for MZ (Table 2).

### 3.2. Apoptosis and Necrosis Time-Course

The treatment of HepG2 cells with MZ induced a continuous time-dependent decrease in the apoptotic signal at the highest concentration (295.7 μM), which was significant after 4 h up to 24 h; at 24 h, the lower dose (2.96 μM) also induced a significant decrease in apoptosis (Figure 2A). Correspondingly, the highest dose induced a time-dependent increase in necrosis, which was statistically significant from 2 h up to 8 h, and decreased at 24 h (Figure 2B). In A549 cells, the lowest dose of MZ significantly decreased apoptosis from 3 h up to 8 h (Figure 2C), but no significant effect on necrosis was observed; MZ treatment at the highest concentration significantly induced necrosis at 24 h (Figure 2D).

Treatment of HepG2 cells with ZOX significantly increased both the apoptotic and necrotic signals constantly over time in almost all three tested concentrations. In addition, the increase in necrotic signals is dose-dependent (Figure 3A,B). In A549 cell line, treatment with ZOX increased apoptosis at the intermediate concentration at early times of the incubation (1, 2 and 4 h), while no effect on necrosis was observed at any concentration tested (Figure 3C,D).

### 3.3. ROS Intracellular Levels

MZ and ZOX affected intracellular ROS production differently in HepG2 and A549 cell lines, as shown in Figure 4. MZ exerted an inverse dose-related increase in intracellular ROS in HepG2 and A549 cells, which was more evident in HepG2 cells but significant only in A549 at the 2.96 μM concentration (Figure 4). ZOX treatment did not markedly affect ROS levels in HepG2, although a slight but statistically significant decrease was observed upon treatment with the highest concentration (463.4 μM). In A549, a dose-related increase in intracellular ROS was noted following ZOX treatment, although it was not significant.

Both cell lines treated with the positive control H_2_O_2_ 100 mM showed significantly higher ROS levels (data not shown).

### 3.4. Comet Assay

A significant increase in % TI was observed in HepG2 cells following treatment with MZ 296 nM after 5 h exposure, whereas no alteration was present following 24 h treatment at any concentration tested (Figure 5).

A statistically significant increase in % TI was observed following treatment with MZ 29.6 μM for 24 h in A549 cells, while no effect was registered after 5 h treatment.

ZOX resulted in an increase in % TI in HepG2 cells at the highest concentration tested (46.3 μM) after 5 h and the effect disappeared after 24 h treatment. No significant effect was observed in A549 treated with ZOX, at both times of observation.

Both cell lines treated with the positive control MMS 1 mM for 30 min showed a significant increase in % TI (data not shown), confirming the robustness of the performed assay. Images acquired by fluorescent microscope are shown in Figure 6.

### 3.5. Micronucleus Test

HepG2 cell line treated with MZ showed a dose-dependent increase in MN+ frequencies compared to the control group, and was statistically significant in concentrations from 29.6 nM to 14.79 μM (Figure 7). MZ also induced a similar increasing trend in MN–, which was significant at the two highest concentrations tested (2.96 and 14.79 μM). In A549, the MN+ dose-dependent increase determined by MZ was significant at the two highest concentrations (2.96 and 29.6 μM), whereas MN− were significantly increased only at 2.96 μM.

ZOX treatment of HepG2 induced a dose-dependent increase in MN+ at the two highest concentrations tested (4.63 and 5.79 μM), whereas in A549 the number of MN+ significantly increased following treatment with all concentrations. No effect on MN− was recorded, neither in HepG2 nor in A549 cells. Images acquired by fluorescent microscope are shown in Figure 8.

### 3.6. Gene Expression

Expression analysis of genes related to apoptosis showed that in both HepG2 and A549 cell lines MZ exerted an inverse dose-related increase in the expression of the pro-apoptotic *BAX* gene, which was more evident and significant only in A549 cells (Figure 9A,B). MZ induced different effects on the anti-apoptotic *BCL2* gene expression in the two cell types, with down-regulation at the two higher concentrations in HepG2 and up-regulation at the two lower concentrations in A549. This led to a significant increase in the *BAX/BCL2* ratio in HepG2 only at the lowest concentration tested.

Treatment with ZOX in HepG2 significantly decreased *BAX* expression and increased *BCL2* expression in A549 at the two higher concentrations. As a result, a significant decrease in the *BAX/BCL2* ratio was observed in both cell lines at all concentrations tested (Figure 9A,B).

As regards the genes involved in oxidative stress regulation, *NRF2* was not affected by MZ and ZOX treatment in HepG2 cells and only the highest MZ dose increased *CYP1A1* expression (Figure 10A); both compounds induced *NRF2* expression in A549 at the two lower or at the 46.3 nM and 4.63 μM concentrations, respectively (Figure 10B). In this last cell line, at doses where the effect was more evident (MZ 2.96 μM and ZOX 4.63 μM), an increase in *CYP1A1* expression was also observed.

Among the genes regulating DNA repair mechanisms, *ERCC1* was significantly up-regulated by ZOX at the middle dose in HepG2, whereas a significant increase was observed in A549 at all concentrations tested with both compounds (Figure 10A,B). *OGG1* was down-regulated by MZ treatment at the two higher concentrations in HepG2 and at 29.6 μM in A549.

## 4. Discussion

The present study confirmed and expanded previous evidence of the toxicity of the fungicide MZ and provided novel evidence of ZOX effects on human liver and lung cell lines, in a range of concentrations starting from those directly used in the field by agricultural workers. Indeed, the in vitro tests selected and included in such a complex approach contributed to identify the different modes of action of MZ and ZOX. Even though both compounds were cytotoxic at the two highest concentrations, MZ appeared to have a more marked effect on HepG2 and A549 cell vitality compared to cell proliferation, as evidenced by the corresponding EC_50_. On the contrary, ZOX significantly affected the cell proliferation of both cell lines, and vitality was affected by ZOX only in HepG2 cells without leading to cytotoxicity. The present results for MZ are in line with previous evidence; in fact, although EC_50_ were not reported, the first significant effect on cell viability after 24 h treatment was observed for 10 μM of MZ in gastric cells [10] and for 40 μM in Caco2 cells [9]. Similar results were observed for gastric cell proliferation after 48 h treatment with 5 μM of MZ [10]. Conversely, a non-monotonic effect on cell vitality and a marked decrease in cell proliferation were observed in murine macrophages [29], thus highlighting a different, pleiotropic effect of MZ. To our knowledge, only one report is available on the in vitro toxicity assessment of ZOX, which investigated cell vitality and proliferation up to 300 nM in V79 cells and showing no significant effect [30]. In the only available publication on HepG2 cells, MZ was tested in the millimolar range [8], thus no comparison is possible with the present results. No data are available for both compounds in A549 cell lines.

The time-course analysis of apoptotic and necrotic processes confirmed the different behaviors of the two fungicides. In HepG2 cells, the marked increase in the necrotic signal induced by the highest concentration of MZ supported the hypothesis that cytotoxicity was due to extensive necrosis of the cells; as confirmation, proliferating and viable cells were decreased at the same concentration. Moreover, BCL2 expression was decreased at the two higher concentrations and the *BAX/BCL2* ratio was increased at the lowest concentrations. Although the 2.96 μM concentration was not cytotoxic, the transcriptional modulation may indicate an early effect. More moderated effects were observed in A549 cells, with necrosis appearing only after 24 h treatment at the highest dose; surprisingly, the apoptotic signal was decreased by the lowest dose, but in this case this may also be suggestive of an early effect. Both *BAX* and *BCL2* were induced by MZ in A549, which may imply a different activating mechanism of *BCL2* not limited to cell survival regulation [31]. These results are supported by available evidence reporting an increase in apoptotic cell numbers upon exposure of lymphocytes to 0.5–5 µg/mL MZ (equal to 0.92–9.24 µM) [12] as well as of gastric cells exposed to 5–10 µM MZ [10], with a concomitant dose-dependent increase in *BAX* and decrease in *BCL2* protein expression in both cells.

ZOX significantly increased apoptosis and necrosis in HepG2 compared to control cells at all tested concentrations and with a constant effect over time, thus supporting the cytotoxicity observed in these cells. In A549, no necrotic effect and only a minor increase in apoptotic signals were recorded, corresponding to a drop in cell proliferation. To our knowledge, this is the first report showing ZOX apoptotic effects in cell models.

Although previous studies showed that MZ may cause mitochondrial dysfunction via ROS generation in different mammalian systems [10,32,33,34,35], this was not confirmed by the present data in such experimental conditions. The isolated ROS increase in A549 at 2.96 µM was probably due to the reduced number of cells at the cytotoxic concentrations, which also affected the detectable amount of intracellular ROS. Consistent with such evidence, the up-regulation of the pro-oxidant transcription factor *NRF2* in this cell line was observed, whereas no effect was recorded in HepG2. Similarly, the expression level of the gene encoding for *CYP1A1*, one of the main P450 enzymes primarily involved in xenobiotics’ metabolism, was up-regulated in A549 at 2.96 µM as well as in HepG2 at 29.6 µM. Overall, this evidence was suggestive of the activation of the aryl hydrocarbon receptor (AhR), which regulates the transcription of *NRF2* and *CYP1A1* [36,37], both involved in the control of the expression of anti-oxidant enzymes that, as last, act as pro-oxidants and lead to the oxygenation of exogenous compounds, thus contributing to the production of ROS. Depending on their balance, cells may undergo to oxidative stress, cell toxicity and finally to death or not [38,39]. Although limited effects on ROS production were observed, cytotoxicity and apoptosis were significantly induced by MZ in both cell lines, thus an imbalance toward CYP1A1 activation may be hypothesized.

ZOX induced a dose-related, but not significant, increase in ROS intracellular levels in A549 cells accompanied by *NRF2* and *CYP1A1* up-regulation; in HepG2 cells, this was not recorded, thus a clearer mechanism involving the oxidative stress response may be depicted in A549 compared to HepG2. To our knowledge, no report up to now has described ZOX’s effects on such endpoints.

Comet assay, as expected, confirmed the capacity of MZ to induce DNA damage. Indeed, several in vitro studies have shown the same effects in human and rat cultured peripheral blood lymphocytes, proposing the prooxidant action of MZ as leading cause of DNA breakage and genotoxicity [11,12,13]. Micronucleus assay confirmed MZ damaging effect, with a dose dependent increase in both MN− and MN+ frequencies in both cell lines, suggestive of a double clastogenic and aneugenic potential. In fact, although the increase in unrepaired clastogenic damage (MN−) was expected due to the MZ mode of action, as confirmed by previous studies, and partially by the present comet assay results, its aneugenic properties have not been described before. Only few studies are available on this topic; as an example, Liu et al. [40] recently showed that a novel dithiocarbamate hybrid molecule was able to inhibit tubulin polymerization in the HepG2 cell line. In addition, Solonesky et al. [41] reported that Zineb, another EBDTC fungicide similar to MZ, caused several spindle apparatus abnormalities in Chinese hamster ovary and HeLa cell lines. The present findings, for the first time, provide evidence that MZ is able to induce aneuploidy in two different human cell lines, probably by interfering with tubulin assembly in the mitotic spindle.

Interestingly, the gene expression of two main regulators of DNA repair systems, i.e., *ERCC1* and *OGG1* involved in nucleotide excision repair (NER) and base excision repair (BER) systems, respectively [42,43], was differently modulated by MZ. This supported the deregulation of 8-oxoG removal as the primary cause of the DNA damage due to MZ exposure.

Indeed, since the alkaline version of the comet assay is also able to measure alkali-labile sites, i.e., AP-sites caused by 8-oxoG, by converting them into breaks under basic conditions [44], and considering that the MN test measures clastogenic unrepaired DNA damage, the present findings provide further evidence of the oxidative mechanism of DNA damage caused by MZ.

Moreover, the present study investigated for the first time the in vitro genotoxic potential of ZOX in human cell lines. As expected, based on the hypothesized mode of action [16,17], ZOX does not seem to induce direct DNA damage, although the transient increase in tail intensity at 5 h in HepG2 cells probably activates DNA repair systems. In fact, cells with damaged DNA can achieve their repair in different time frames, from few minutes to hours, depending on the severity and/or type of damage, detected by comet assay [45,46]. The dose-related increase in MN+, but not in MN−, especially in A549 cells, suggests a possible aneugenic potential. Indeed, the well-known ZOX interaction with the β-tubulin of the mitotic spindle [47,48], rather than direct interaction with DNA, may underlie the significant increase in chromosome malsegregation.

The up-regulation of *ERCC1* gene in HepG2 cell line treated with ZOX 463 nM and in A549 at all tested doses, and the lack of variation in OGG1 expression supported the hypothesis of an efficient NER system activation preventing the DNA damage, and it confirmed that the increase induced by ZOX may be related to impingement on the β-tubulin structure.

Overall, the present in vitro approach evidenced a higher toxicity of MZ compared to ZOX and a less responsiveness of A549 cells with respect to HepG2 cells. This last piece of evidence is in agreement with previous observations, which compared cytotoxicity and genotoxicity of several herbicides [49] and it might be explained by the lower subset of cytochrome P450 enzymes expressed in the lung compared to the complete CYPs milieu expressed in the liver [50].

## 5. Conclusions

The toxicological and genotoxic potential of MZ and ZOX was comparatively assessed by an in vitro test battery in HepG2 and A549 cell lines treated at concentrations derived from those used in the field by agricultural workers. Both fungicides proved to be cytotoxic but with different mechanisms: MZ possibly triggered an AhR oxidative stress response, leading to oxidative DNA damage and consequently, to cell death by apoptosis with early secondary necrosis phenotypes. The results for ZOX suggest the involvement of other mechanisms of programmed cell death not involving ROS production and apoptosis.

Both fungicides were found to be genotoxic in both cell lines; MZ showed an increase in direct DNA damage and in the dose-dependent MN− and MN+ frequencies in both cell lines. These results suggested the dual clastogenic and aneugenic potential of MZ, which supports the limitations placed on its use in several countries in recent years. ZOX did not induce any direct DNA damage, probably due to the activation of NER repair system, but caused a dose-dependent increase in the MN+ frequency in both cell lines, which indicates its indirect aneugenic potential, possibly through the interaction with the β-tubulin in the mitotic spindle, leading to chromosome malsegregation.

It is known that aneuploidy represents an important chromosomal alteration that may be associated with primary events of carcinogenesis [51]. Indeed, the present data provides new evidence of ZOX’s toxicity, and raises concerns regarding its use.

## Figures and Tables

**Figure 1 ijerph-18-08591-f001:**
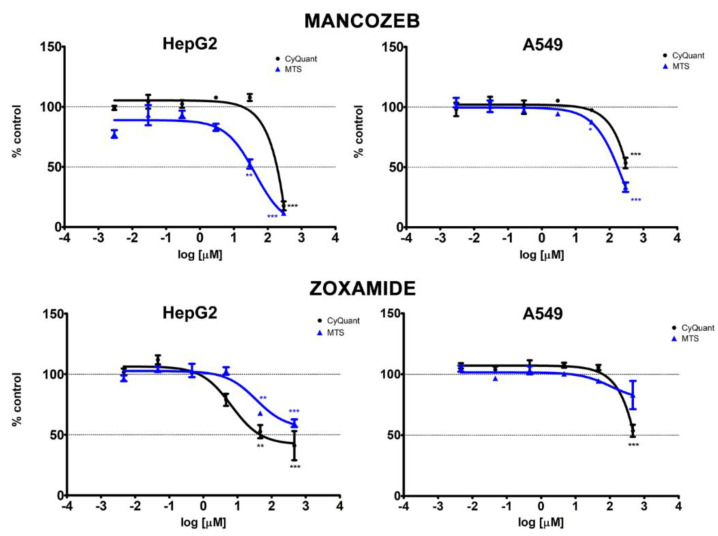
Dose–response curves of CyQUANT (black lines) and MTS (blue lines) assays in HepG2 and A549 cells treated with MZ and ZOX for 24 h. Values are means ± standard error of the mean (SEM) of three independent experiments normalized to the solvent control (0.8% DMSO for MZ; 0.2% DMSO for ZOX). Statistical significance is indicated by asterisks: * *p* < 0.05; ** *p* < 0.01; *** *p* < 0.001.

**Figure 2 ijerph-18-08591-f002:**
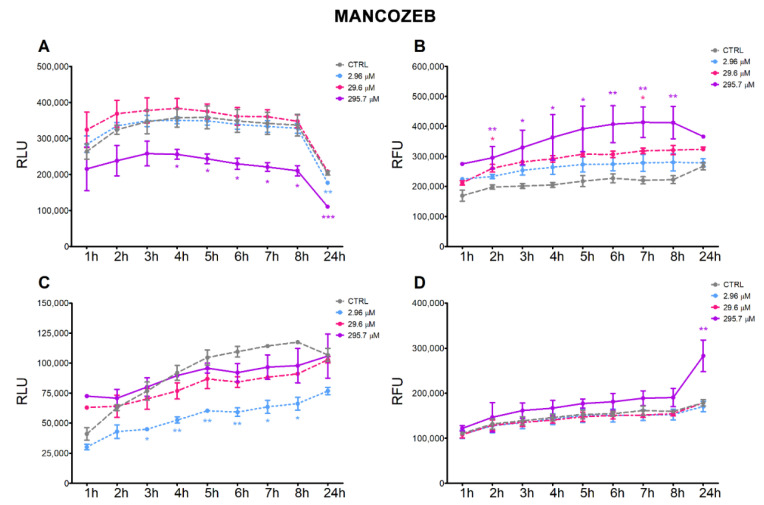
Apoptotic (RLU) and necrotic (RFU) signals evaluated by Annexin V assay in HepG2 (**A**,**B**) and A549 (**C**,**D**) cells treated with MZ or solvent control (0.8% DMSO) for 24 h. Values are mean fold-change ± SEM of three independent experiments. Statistical significance is indicated by asterisks: * *p* < 0.05; ** *p* < 0.01; *** *p* < 0.001.

**Figure 3 ijerph-18-08591-f003:**
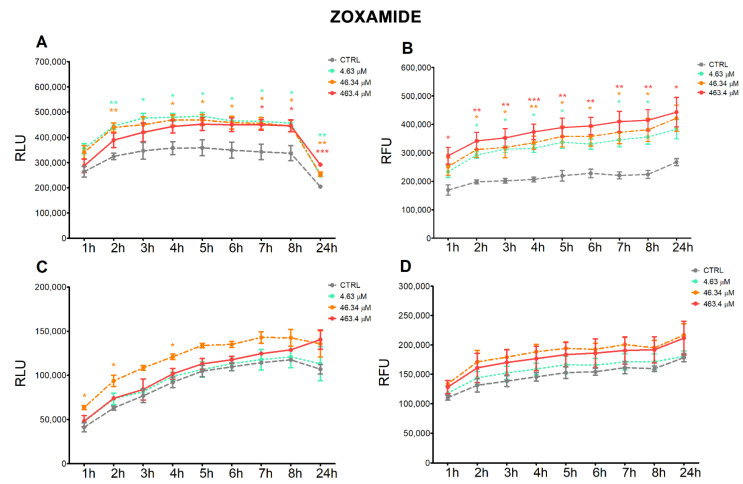
Apoptotic (RLU) and necrotic (RFU) signals evaluated by Annexin V assay in HepG2 (**A**,**B**) and A549 (**C**,**D**) cells treated with ZOX or solvent control (0.2% DMSO) for 24 h. Values are mean fold-change ± SEM of three independent experiments. Statistical significance is indicated by asterisks: * *p* < 0.05; ** *p* < 0.01; *** *p* < 0.001.

**Figure 4 ijerph-18-08591-f004:**
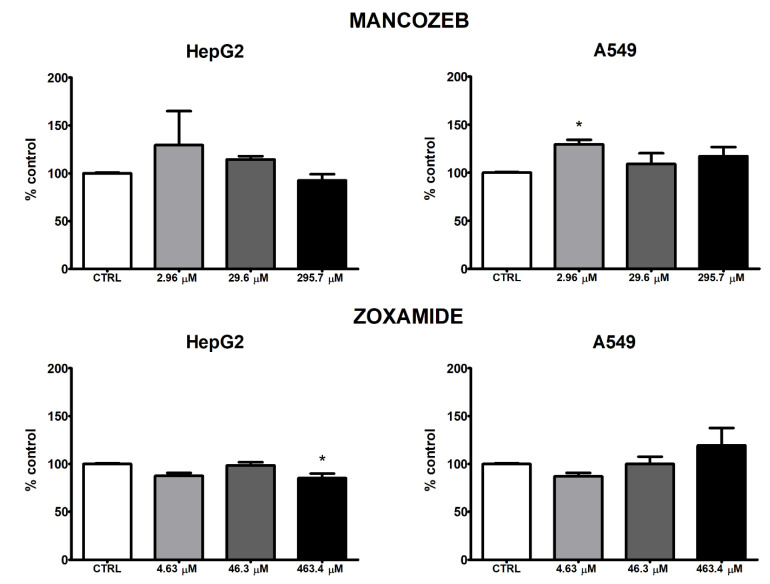
ROS production in HepG2 and A549 cells treated with MZ and ZOX for 24 h. Values are means ± SEM of three independent experiments, normalized to the solvent control (0.8% DMSO for MZ; 0.2% DMSO for ZOX) set at 100%. Statistical significance is indicated by asterisks: * *p* < 0.05.

**Figure 5 ijerph-18-08591-f005:**
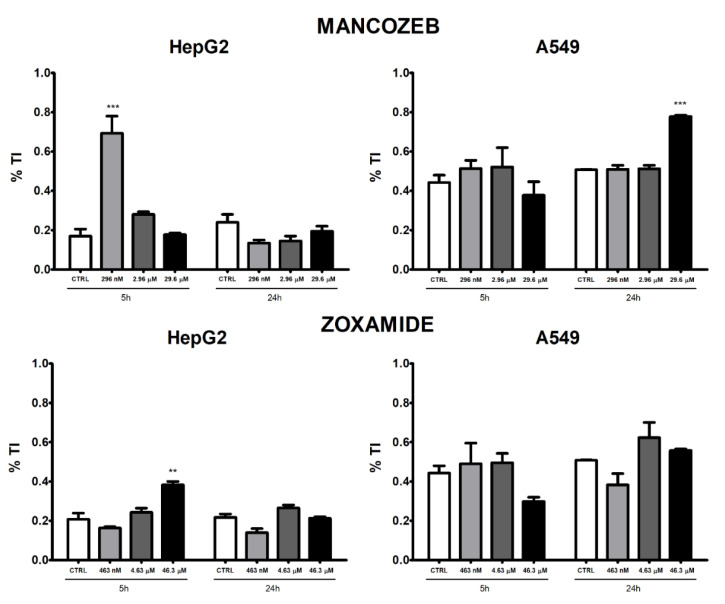
DNA damage measured as percentage of tail intensity (% TI) by comet assay in HepG2 and A549 cells treated with MZ and ZOX or medium alone as control for 24 h. Values are means ± SEM of three independent experiments. Statistical significance is indicated by asterisks: ** *p* < 0.01; *** *p* < 0.001.

**Figure 6 ijerph-18-08591-f006:**
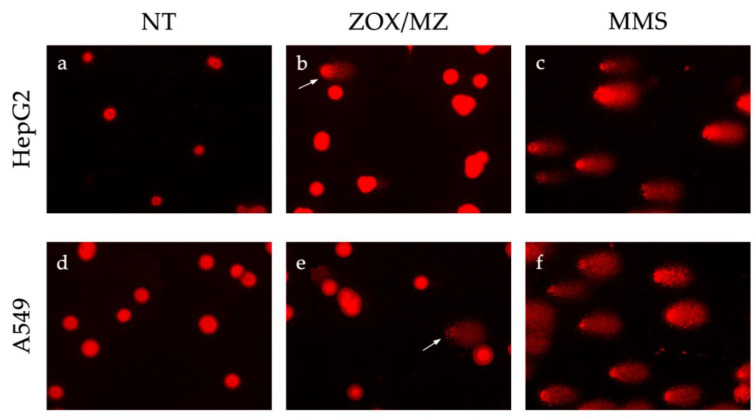
Fluorescence microscope images of HepG2 (**a**–**c**) and A549 (**d**–**f**) nucleoids. Control cells (**a**,**d**); HepG2 treated with ZM 46.34 μM for 5 h (**b**); A549 treated with MZ 29.6 μM for 24 h (**e**); cells treated for 30 min with MMS 1 mM as positive controls (**c**,**f**). White arrows show the nucleoids with damaged DNA or “comets”.

**Figure 7 ijerph-18-08591-f007:**
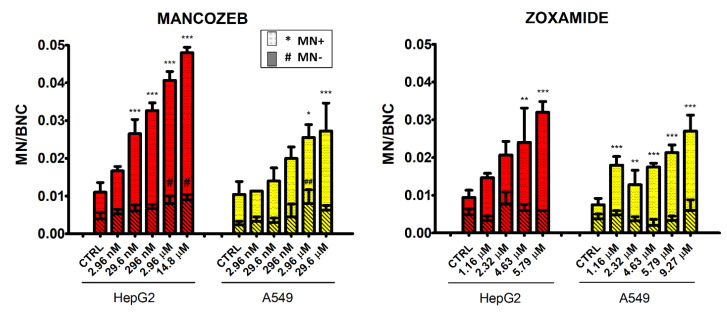
Frequencies of MNi counted in 500 binucleated HepG2 (red) and A549 (yellow) cells after 24 h of treatment with MZ, ZOX or medium alone as control. Bars with striped pattern represent the number of CREST-negative micronuclei (MN−), filled bars represent the number of CREST-positive micronuclei (MN+). Values are means ± SEM of three independent experiments. Statistical significance is indicated by asterisks for MN+ comparisons and by hashtags for MN-comparisons: *, ^#^ statistically significant (*p* < 0.05); **, ^##^ (*p* < 0.01); *** (*p* < 0.001).

**Figure 8 ijerph-18-08591-f008:**
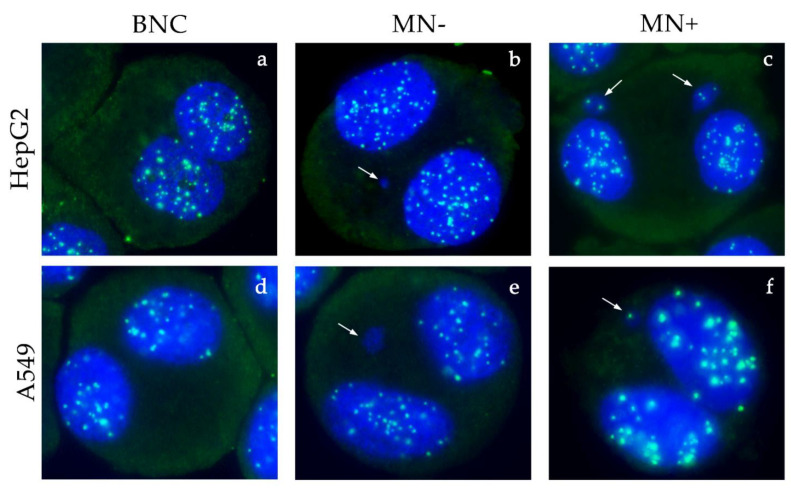
Fluorescence microscope images of HepG2 (**a**–**c**) and A549 (**d**–**f**) binucleated cells stained with DAPI and CREST antibodies. Normal binucleated cells in control group (**a**,**d**); CREST-negative micronuclei (MN−) in HepG2 treated with MZ 2.96 μM (**b**) and A549 treated with MZ 29.6 μM (**e**); double CREST-positive micronuclei (MN+) in HepG2 treated with ZM 4.63 μM (**c**); CREST-positive micronucleus (MN+) in A549 treated with ZM 9.27 μM (**f**). Micronuclei are indicated by the white arrows.

**Figure 9 ijerph-18-08591-f009:**
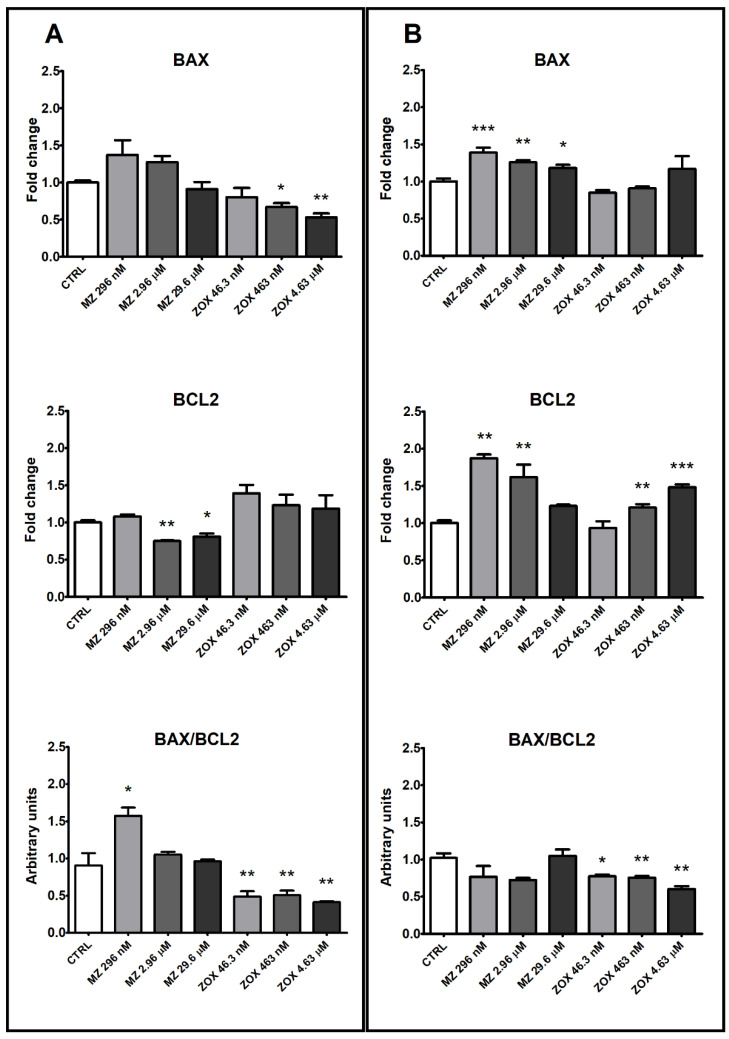
BAX and BCL2 gene expression analysis by real-time PCR in HepG2 (**A**) and A549 (**B**) cells following treatment with MZ, ZOX or medium alone as control for 24 h. Data are mean ΔΔCt values ± SEM of three biological replica, with control samples as calibrators and GAPDH as the reference gene. Statistical significance is indicated by asterisks: * *p* < 0.05; ** *p* < 0.01; *** *p* < 0.001.

**Figure 10 ijerph-18-08591-f010:**
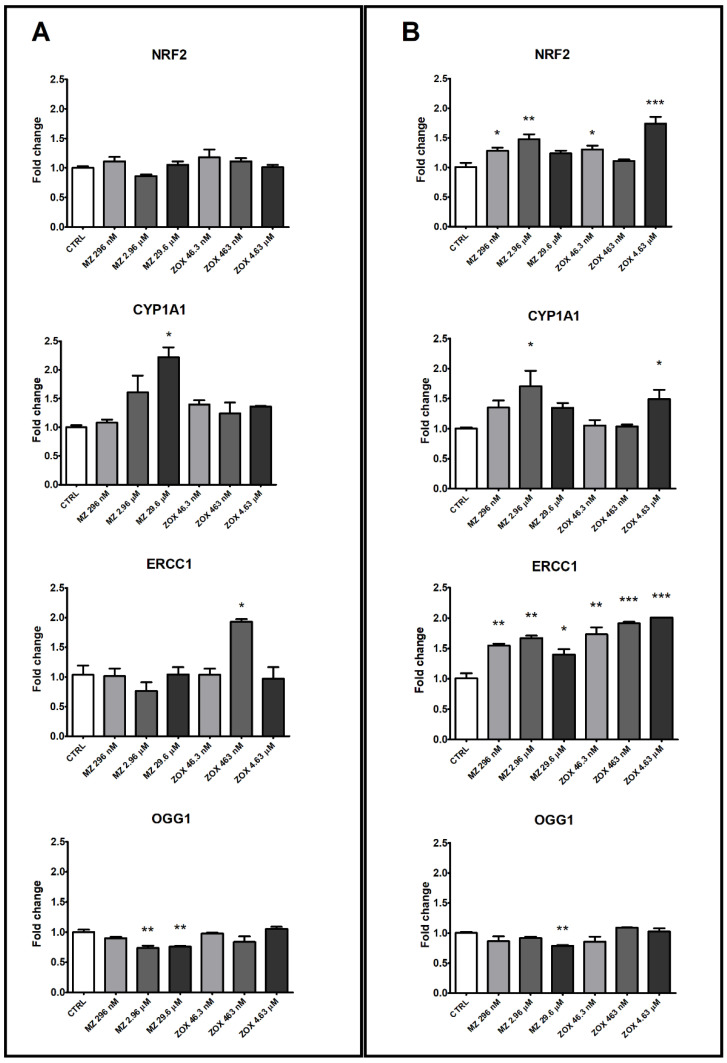
NRF2, CYP1A1, ERCC1 and OGG1 gene expression analysis by real-time PCR in HepG2 (**A**) and A549 (**B**) cells following treatment with MZ, ZOX or medium alone as control for 24 h. Data are mean ΔΔCt values ± SEM of three biological replica, with control samples as calibrators and GAPDH as the reference gene. Statistical significance is indicated by asterisks: * *p* < 0.05; ** *p* < 0.01; *** *p* < 0.001.

**Table 1 ijerph-18-08591-t001:** Forward and reverse sequences of primers used in real-time PCR.

Gene	qPCR Primers (5′-3′)	
*GAPDH*	fw:	ACTCCTCCACCTTTGACGCT
	rev:	CTTCAAGGGGTCTACATGGC
*BAX*	fw:	GTCTTTTTCCGAGTGGCAGC
	rev:	GACAGGGACATCAGTCGCTT
*BCL2*	fw:	CTTTGAGTTCGGTGGGGTCA
	rev:	GGGCCGTACAGTTCCACAAA
*NRF2*	fw:	ACAAGATGGGCTGCTGCACTGG
	rev:	TCCCCGAGGAACCCGCTGAAAA
*CYP1A1*	fw:	CCCCCACAGCACAACAAGAG
	rev:	GGGTGAGAAACCGTTCAGGT
*ERCC1*	fw:	GGCTCGAGAAAGACAGGCTCC
	rev:	CATATTCGGCGTAGGTCTGAGG
*OGG1*	fw:	GCCTGATGGCCCTAGACAAG
	rev:	GCACTGAACAGCACCGCTT

**Table 2 ijerph-18-08591-t002:** EC_10_, EC_20_ and EC_50_ expressed in µM concentrations, calculated from dose–response curve of CyQuant and MTS assays in HepG2 and A549 cells treated with MZ and ZOX for 24 h. NA (not available).

Chemical	Test	HepG2			A549		
		EC_10_ (95% CI)	EC_20_ (95% CI)	EC_50_ (95% CI)	EC_10_ (95% CI)	EC_20_ (95% CI)	EC_50_ (95% CI)
Mancozeb	MTS	3.61 (<2.96 × 10^−3^–8.04)	8.67 (1.01–16.34)	38.73 (19.11–58.35)	22.89 (7.82–37.96)	47.13 (24.66–69.61)	162.03 (121.68–202.38)
	CyQUANT	142.43 (<2.96 × 10^−3^–677.98)	166.45 (<2.96 × 10^−3^–658.65)	217.96 (<2.96 × 10^−3^–560.64)	69.39 (<2.96 × 10^−3^–181.19)	121.89 (0.16–243.59)	319.24 (251.16–387.32)
Zoxamide	MTS	14.45 (<4.63 × 10^−3^–41.14)	61.87 (<4.63 × 10^−3^–135.78)	NA	NA	NA	NA
	CyQUANT	0.75 (<4.63 × 10^−3^–2.09)	1.79 (<4.63 × 10^−3^–4.09)	7.90 (<4.63 × 10^−3^–18.50)	200.57 (<4.63 × 10^−3^–589.47)	274.82 (<4.63 × 10^−3^–608.30)	470.84 (420.00–521.68)

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
