# Peer review of "Toxicological Comparison of Mancozeb and Zoxamide Fungicides at Environmentally Relevant Concentrations by an In Vitro Approach"

_ijerph, 2021, doi:10.3390/ijerph18168591_

Round 1

Reviewer 1 Report

Congratulations on an interesting and very elaborate publication in terms of markings made. The methodology is adequate to the objectives of the work. I think it fully deserves to be published in its present form.

Author Response

REVIEWER #1

Congratulations on an interesting and very elaborate publication in terms of markings made. The methodology is adequateto the objectives of the work. I think it fully deserves to be published in its present form.

Response

We really thank the Reviewer for the kind comment and for the appreciation for our manuscript.

REVIEWER #2

The manuscript titled: “Toxicological comparison of Mancozeb and Zoxamide fungicides at environmentally relevant concentrations by an in vitro approach” shows an interesting article depicting a comparison of Mancozeb and Zoxamide fungicides regarding their potential cytotoxic effects on human cancer cell lines. The manuscript is well written, scientifically sound, and the results are well explained. However, I think the authors should justify better using cancer cell lines to evaluate cytotoxic potential because, as indicated in the manuscript. I suggest minor comments for this manuscript to be suitable for publication on IJERPH.  

ABSTRACT 

  1. Please use italics when referring to gene names. This applies to the entire document.

The correction has been done as suggested.

  1. Line 20: What were the highest tested concentrations? Please indicate the values.

The concentrations have been indicated in the text as suggested.

  1. Line 20: By “cytotoxic,” do the authors mean cell viability < 80 %? Please complete. 

Yes, “cytotoxic” means that cell viability/proliferation is less than 70-80% compared to control cells (as also indicated at line 157 of the revised manuscript) but we did not specify the exact percentage due to the word limit of the abstract; however, both compounds decreased cell viability/proliferation by over 50% at the highest concentration tested. A complete description of the cytotoxic effects is present in the results section.

KEYWORDS 

  1. Please alphabetically re-arrange keywords. 

We re-arranged the keywords as requested.

INTRODUCTION 

  1. Lines 71-72: Please explain how cancer cell lines sustain the evaluation of cytotoxicity of these fungicides on normal cell lines? 

We are aware that normal cell lines better recapitulate human physiology, however both HepG2 and A549 are widely implied in toxicological studies and, at least HepG2, in high throughput assays in the ToxCast program of EPA; for these reasons, they have been used to allow a broader and sound comparison with available data. Such consideration has also been added in the text.

MATERIALS AND METHODS 

  1. Lines 81-82: Please refer to ATCC on how to report these cells lines properly. For instance, I have attached a link from ATCC where the product sheet in the “Material Citation” section indicates that HepG2 cells should be cited as Hep G2 [HEPG2] (ATCC HB-8065). Please double-check this for the assayed cells lines in this manuscript 

https://www.atcc.org/products/hb-8065 (download the “Product Sheet” file). 

The correct citations have been included as suggested

  1. Line 101: Did the authors mean “six to ten-fold”?

No, we meant six concentrations, each diluted ten-fold with respect to the previous one.

  1. Lines 110-111: How many replicates were considered for each independent experiment? Please complete this information for the entire document.

Thank you for the comment; indeed, this information is already present in the entire document and specifically: at line 112 (current reviewed manuscript) it is indicated that assays were performed in triplicate and, at line 118, that each assay was performed three times in independent experiments. For the other assays, we indicated that they were performed in duplicated wells (line 129 for Apoptosis/Necrosis assay; line 145 for ROS assay) or in triplicated dishes (line 160 for Comet assay) and repeated the assay in three independent experiments (line 134 for Apoptosis/Necrosis assay; 147 for ROS assay; line 180 for Comet Assay). For micronucleus assay and cell treatment for gene expression analysis, three independent experiments were performed (lines 210 and 218, respectively).

  1. Did the authors use black 96-well plates for the ROS quantification? Please add this information.

No, we did not use black 96-well plates because it was not strictly requested by the manufacturer’s protocol, indicating black or clear plates. Otherwise, for Apoptosis/Necrotic assays we used white 96-well plates as specifically requested by the protocol, as indicated in the text.

  1. Line 156: Please add the trypsin concentration. Did the trypsin contain EDTA?

We added the trypsin concentration (0.25%) and the supplier. The trypsin did not contain EDTA.

  1. Line 157: Please report speeds in gravity as rpm depends on the centrifuge. Moreover, add the temperature used.The same comment applies to the entire document.

We changed speeds from rpm to rcf where indicated, except for the centrifugation at the cytospin since the instrument reports only rpm.

  1. Section 2.7: Please write all gene names using italics. 

The names have been changed as indicated by the Reviewer.

RESULTS AND DISCUSSION 

  1. Figure 2-5: I think it would be easier to follow the figures if you add legends for each one about what compound is being tested, without the need to necessarily address the figure caption (for instance, as presented in Fig. 1, Fig. 6, or Fig. 10).

As suggested by the Reviewer, we added legends to the indicated figures; for consistency, we added legends also to figures 1 and 7.

  1. This is only a suggestion: I believe the authors could add a mechanistic figure summarizing all the obtained results. I think this would enormously benefit the manuscript. For example, please refer to these articles: 

Figure 6, https://www.sciencedirect.com/science/article/pii/S0278691521001526

Figure 7, https://www.sciencedirect.com/science/article/pii/S1756464620304941

Figure 10, https://www.sciencedirect.com/science/article/pii/S0963996921002179

We really thank the Reviewer for this suggestion; however, Mancozeb and Zoxamide exerted different mechanisms and to a different extent in the two cell lines: in this respect, one single figure could not have been sufficient to summarize all these complex effects and more than one figure could have weighted the manuscript down. Thus, we decided to leave the description in the text.

REVIEWER #3

This manuscript aims to develop an in vitro testing approach, to compare Mancozeb and Zoxamide and to evaluate the actual validity of Mancozeb substitution, in the agronomical protocols.

The manuscript provides an organic comparison, evaluating the cytotoxic, apoptotic and genetoxic potential of these fungicides, at environmentally relevant concentrations. Even though this article is densely organized and based on recent and well-synthetized data, the are aspects to be mentioned, to make the article fully readable. For these reasons, the manuscript requires minor changes.

Please find below an enumerated list of comments on my review of the manuscript:

LINE 51:

Mancozeb and organic pollutants’ effects on endocrine system are also reported in other studies (see, for reference: Exposure to persistent organic pollutants during tooth formation: molecular mechanisms and clinical findings – 2020), highlighting their role in alterations to the endocrine and developmental systems, at environmental exposure levels.

Thank you for the suggestion. Unfortunately, our institution has no access to the journal where the reference is published, so we couldn’t evaluate the suggested paper. Anyway, we added in the text a sentence on Mancozeb effects as endocrine disruptor (A systematic review of Mancozeb as a reproductive and developmental hazard – 2017), as supporting evidence.

LINE 53:

The manuscript may benefit from highlighting also Mancozeb gonadal toxicity, specifically on female reproductive system. Due to its persistence and versatile profile, Mancozeb exposure is linked to several female reproductive outcomes, as confirmed by different in vivo and in vitro studies (see, for reference: Protective effects of resveratrol against mancozeb induced apoptosis damage in mouse oocytes – 2017; Transcriptional landscape of mouse-aged ovaries reveals a unique set of non-coding RNAs associated with physiological and environmental ovarian dysfunctions – 2018; Mancozeb exposure during development and lactation periods results in decreased oocyte maturation, fertilization rates, and implantation in the first-generation mice pups: Protective effect of vitamins E and C; Association between Female Reproductive Health and Mancozeb: Systematic Review of Experimental Models – 2020). In this context, Mancozeb plays a pleiotropic role, exerting its toxicity on overall health.

Thank you for the interesting information. Nevertheless, in our opinion, the suggested references do not properly fit the manuscript which is mainly focused on liver and pulmonary adverse effects potentially occurring to agricultural workers directly exposed to agrochemicals. Anyway, as indicated in the previous answer, we added the information on Mancozeb developmental and reproductive adverse effects in the text.

MATERIAL AND METHODS:

As regards this section, the methodology design was rigorous and appropriately implemented within the study.

RESULTS:

Also this section is well organized and densely presented, based on well-synthetized data.

In this context, Mancozeb and Zoxamide are epigenetic hazards and powerful environmental pollutants, which interact with a health phenotype, changing directly or indirectly its inner molecular and cellular balances. To this aim, it will be useful to analyze in the future also their potential on the epigenetic modifications.

Thank you for the suggestion. Indeed, it is our intention to further evaluate such aspect.

Overall, the manuscript requires major changes (as mentioned). I would accept the manuscript, if the comments are addressed properly

Reviewer 2 Report

The manuscript titled: “Toxicological comparison of Mancozeb and Zoxamide fungicides at environmentally relevant concentrations by an in vitro approach” shows an interesting article depicting a comparison of Mancozeb and Zoxamide fungicides regarding their potential cytotoxic effects on human cancer cell lines. The manuscript is well written, scientifically sound, and the results are well explained. However, I think the authors should justify better using cancer cell lines to evaluate cytotoxic potential because, as indicated in the manuscript. I suggest minor comments for this manuscript to be suitable for publication on IJERPH.  

ABSTRACT

  1. Please use italics when referring to gene names. This applies to the entire document.
  2. Line 20: What were the highest tested concentrations? Please indicate the values.
  3. Line 20: By “cytotoxic,” do the authors mean cell viability < 80 %? Please complete.

KEYWORDS

  1. Please alphabetically re-arrange keywords.

INTRODUCTION

  1. Lines 71-72: Please explain how cancer cell lines sustain the evaluation of cytotoxicity of these fungicides on normal cell lines?

MATERIALS AND METHODS

  1. Lines 81-82: Please refer to ATCC on how to report these cells lines properly. For instance, I have attached a link from ATCC where the product sheet in the “Material Citation” section indicates that HepG2 cells should be cited as Hep G2 [HEPG2] (ATCC HB-8065). Please double-check this for the assayed cells lines in this manuscript

https://www.atcc.org/products/hb-8065 (download the “Product Sheet” file).

  1. Line 101: Did the authors mean “six to ten-fold”?
  2. Lines 110-111: How many replicates were considered for each independent experiment? Please complete this information for the entire document.
  3. Did the authors use black 96-well plates for the ROS quantification? Please add this information.
  4. Line 156: Please add the trypsin concentration. Did the trypsin contain EDTA?
  5. Line 157: Please report speeds in gravity as rpm depends on the centrifuge. Moreover, add the temperature used. The same comment applies to the entire document.
  6. Section 2.7: Please write all gene names using italics.

RESULTS AND DISCUSSION

  1. Figure 2-5: I think it would be easier to follow the figures if you add legends for each one about what compound is being tested, without the need to necessarily address the figure caption (for instance, as presented in Fig. 1, Fig. 6, or Fig. 10).
  2. This is only a suggestion: I believe the authors could add a mechanistic figure summarizing all the obtained results. I think this would enormously benefit the manuscript. For example, please refer to these articles:

Figure 6, https://www.sciencedirect.com/science/article/pii/S0278691521001526

Figure 7, https://www.sciencedirect.com/science/article/pii/S1756464620304941

Figure 10, https://www.sciencedirect.com/science/article/pii/S0963996921002179

Author Response

REVIEWER #2

The manuscript titled: “Toxicological comparison of Mancozeb and Zoxamide fungicides at environmentally relevant concentrations by an in vitro approach” shows an interesting article depicting a comparison of Mancozeb and Zoxamide fungicides regarding their potential cytotoxic effects on human cancer cell lines. The manuscript is well written, scientifically sound, and the results are well explained. However, I think the authors should justify better using cancer cell lines to evaluate cytotoxic potential because, as indicated in the manuscript. I suggest minor comments for this manuscript to be suitable for publication on IJERPH.  

ABSTRACT 

  1. Please use italics when referring to gene names. This applies to the entire document.

The correction has been done as suggested.

  1. Line 20: What were the highest tested concentrations? Please indicate the values.

The concentrations have been indicated in the text as suggested.

  1. Line 20: By “cytotoxic,” do the authors mean cell viability < 80 %? Please complete. 

Yes, “cytotoxic” means that cell viability/proliferation is less than 70-80% compared to control cells (as also indicated at line 157 of the revised manuscript) but we did not specify the exact percentage due to the word limit of the abstract; however, both compounds decreased cell viability/proliferation by over 50% at the highest concentration tested. A complete description of the cytotoxic effects is present in the results section.

KEYWORDS 

  1. Please alphabetically re-arrange keywords. 

We re-arranged the keywords as requested.

INTRODUCTION 

  1. Lines 71-72: Please explain how cancer cell lines sustain the evaluation of cytotoxicity of these fungicides on normal cell lines? 

We are aware that normal cell lines better recapitulate human physiology, however both HepG2 and A549 are widely implied in toxicological studies and, at least HepG2, in high throughput assays in the ToxCast program of EPA; for these reasons, they have been used to allow a broader and sound comparison with available data. Such consideration has also been added in the text.

MATERIALS AND METHODS 

  1. Lines 81-82: Please refer to ATCC on how to report these cells lines properly. For instance, I have attached a link from ATCC where the product sheet in the “Material Citation” section indicates that HepG2 cells should be cited as Hep G2 [HEPG2] (ATCC HB-8065). Please double-check this for the assayed cells lines in this manuscript 

https://www.atcc.org/products/hb-8065 (download the “Product Sheet” file). 

The correct citations have been included as suggested

  1. Line 101: Did the authors mean “six to ten-fold”?

No, we meant six concentrations, each diluted ten-fold with respect to the previous one.

  1. Lines 110-111: How many replicates were considered for each independent experiment? Please complete this information for the entire document.

Thank you for the comment; indeed, this information is already present in the entire document and specifically: at line 112 (current reviewed manuscript) it is indicated that assays were performed in triplicate and, at line 118, that each assay was performed three times in independent experiments. For the other assays, we indicated that they were performed in duplicated wells (line 129 for Apoptosis/Necrosis assay; line 145 for ROS assay) or in triplicated dishes (line 160 for Comet assay) and repeated the assay in three independent experiments (line 134 for Apoptosis/Necrosis assay; 147 for ROS assay; line 180 for Comet Assay). For micronucleus assay and cell treatment for gene expression analysis, three independent experiments were performed (lines 210 and 218, respectively).

  1. Did the authors use black 96-well plates for the ROS quantification? Please add this information.

No, we did not use black 96-well plates because it was not strictly requested by the manufacturer’s protocol, indicating black or clear plates. Otherwise, for Apoptosis/Necrotic assays we used white 96-well plates as specifically requested by the protocol, as indicated in the text.

  1. Line 156: Please add the trypsin concentration. Did the trypsin contain EDTA?

We added the trypsin concentration (0.25%) and the supplier. The trypsin did not contain EDTA.

  1. Line 157: Please report speeds in gravity as rpm depends on the centrifuge. Moreover, add the temperature used.The same comment applies to the entire document.

We changed speeds from rpm to rcf where indicated, except for the centrifugation at the cytospin since the instrument reports only rpm.

  1. Section 2.7: Please write all gene names using italics. 

The names have been changed as indicated by the Reviewer.

RESULTS AND DISCUSSION 

  1. Figure 2-5: I think it would be easier to follow the figures if you add legends for each one about what compound is being tested, without the need to necessarily address the figure caption (for instance, as presented in Fig. 1, Fig. 6, or Fig. 10).

As suggested by the Reviewer, we added legends to the indicated figures; for consistency, we added legends also to figures 1 and 7.

  1. This is only a suggestion: I believe the authors could add a mechanistic figure summarizing all the obtained results. I think this would enormously benefit the manuscript. For example, please refer to these articles: 

Figure 6, https://www.sciencedirect.com/science/article/pii/S0278691521001526

Figure 7, https://www.sciencedirect.com/science/article/pii/S1756464620304941

Figure 10, https://www.sciencedirect.com/science/article/pii/S0963996921002179

We really thank the Reviewer for this suggestion; however, Mancozeb and Zoxamide exerted different mechanisms and to a different extent in the two cell lines: in this respect, one single figure could not have been sufficient to summarize all these complex effects and more than one figure could have weighted the manuscript down. Thus, we decided to leave the description in the text.

Reviewer 3 Report

This manuscript aims to develop an in vitro testing approach, to compare Mancozeb and Zoxamide and to evaluate the actual validity of Mancozeb substitution, in the agronomical protocols.

The manuscript provides an organic comparison, evaluating the cytotoxic, apoptotic and genetoxic potential of these fungicides, at environmentally relevant concentrations. Even though this article is densely organized and based on recent and well-synthetized data, the are aspects to be mentioned, to make the article fully readable. For these reasons, the manuscript requires minor changes.

Please find below an enumerated list of comments on my review of the manuscript:

LINE 51:

Mancozeb and organic pollutants’ effects on endocrine system are also reported in other studies (see, for reference: Exposure to persistent organic pollutants during tooth formation: molecular mechanisms and clinical findings – 2020), highlighting their role in alterations to the endocrine and developmental systems, at environmental exposure levels.

LINE 53:

The manuscript may benefit from highlighting also Mancozeb gonadal toxicity, specifically on female reproductive system. Due to its persistence and versatile profile, Mancozeb exposure is linked to several female reproductive outcomes, as confirmed by different in vivo and in vitro studies (see, for reference: Protective effects of resveratrol against mancozeb induced apoptosis damage in mouse oocytes – 2017; Transcriptional landscape of mouse-aged ovaries reveals a unique set of non-coding RNAs associated with physiological and environmental ovarian dysfunctions – 2018; Mancozeb exposure during development and lactation periods results in decreased oocyte maturation, fertilization rates, and implantation in the first-generation mice pups: Protective effect of vitamins E and C; Association between Female Reproductive Health and Mancozeb: Systematic Review of Experimental Models – 2020). In this context, Mancozeb plays a pleiotropic role, exerting its toxicity on overall health.

MATERIAL AND METHODS:

As regards this section, the methodology design was rigorous and appropriately implemented within the study.

RESULTS:

Also this section is well organized and densely presented, based on well-synthetized data.

In this context, Mancozeb and Zoxamide are epigenetic hazards and powerful environmental pollutants, which interact with a health phenotype, changing directly or indirectly its inner molecular and cellular balances. To this aim, it will be useful to analyze in the future also their potential on the epigenetic modifications. Overall, the manuscript requires major changes (as mentioned). I would accept the manuscript, if the comments are addressed properly.

Author Response

REVIEWER #3

This manuscript aims to develop an in vitro testing approach, to compare Mancozeb and Zoxamide and to evaluate the actual validity of Mancozeb substitution, in the agronomical protocols.

The manuscript provides an organic comparison, evaluating the cytotoxic, apoptotic and genetoxic potential of these fungicides, at environmentally relevant concentrations. Even though this article is densely organized and based on recent and well-synthetized data, the are aspects to be mentioned, to make the article fully readable. For these reasons, the manuscript requires minor changes.

Please find below an enumerated list of comments on my review of the manuscript:

LINE 51:

Mancozeb and organic pollutants’ effects on endocrine system are also reported in other studies (see, for reference: Exposure to persistent organic pollutants during tooth formation: molecular mechanisms and clinical findings – 2020), highlighting their role in alterations to the endocrine and developmental systems, at environmental exposure levels.

Thank you for the suggestion. Unfortunately, our institution has no access to the journal where the reference is published, so we couldn’t evaluate the suggested paper. Anyway, we added in the text a sentence on Mancozeb effects as endocrine disruptor (A systematic review of Mancozeb as a reproductive and developmental hazard – 2017), as supporting evidence.

LINE 53:

The manuscript may benefit from highlighting also Mancozeb gonadal toxicity, specifically on female reproductive system. Due to its persistence and versatile profile, Mancozeb exposure is linked to several female reproductive outcomes, as confirmed by different in vivo and in vitro studies (see, for reference: Protective effects of resveratrol against mancozeb induced apoptosis damage in mouse oocytes – 2017; Transcriptional landscape of mouse-aged ovaries reveals a unique set of non-coding RNAs associated with physiological and environmental ovarian dysfunctions – 2018; Mancozeb exposure during development and lactation periods results in decreased oocyte maturation, fertilization rates, and implantation in the first-generation mice pups: Protective effect of vitamins E and C; Association between Female Reproductive Health and Mancozeb: Systematic Review of Experimental Models – 2020). In this context, Mancozeb plays a pleiotropic role, exerting its toxicity on overall health.

Thank you for the interesting information. Nevertheless, in our opinion, the suggested references do not properly fit the manuscript which is mainly focused on liver and pulmonary adverse effects potentially occurring to agricultural workers directly exposed to agrochemicals. Anyway, as indicated in the previous answer, we added the information on Mancozeb developmental and reproductive adverse effects in the text.

MATERIAL AND METHODS:

As regards this section, the methodology design was rigorous and appropriately implemented within the study.

RESULTS:

Also this section is well organized and densely presented, based on well-synthetized data.

In this context, Mancozeb and Zoxamide are epigenetic hazards and powerful environmental pollutants, which interact with a health phenotype, changing directly or indirectly its inner molecular and cellular balances. To this aim, it will be useful to analyze in the future also their potential on the epigenetic modifications.

Thank you for the suggestion. Indeed, it is our intention to further evaluate such aspect.

Overall, the manuscript requires major changes (as mentioned). I would accept the manuscript, if the comments are addressed properly